# The ubiquitin-like modifier FAT10 covalently modifies HUWE1 and strengthens the interaction of AMBRA1 and HUWE1

**Stefanie Mueller**[1,2], **Johanna Bialas**[1,2], **Stella Ryu**[1,2], **Nicola Catone**[1], **Annette Aichem**[1,2]*

**1** Biotechnology Institute Thurgau at the University of Konstanz, Kreuzlingen, Switzerland, **2** Division of Immunology, Department of Biology, University of Konstanz, Kontstanz, Germany

* Annette.Aichem@bitg.ch

**Data Availability Statement:** All relevant data are within the paper and its Supporting Information files.

## Abstract

The ubiquitin-like modifier FAT10 is highly upregulated under inflammatory conditions and targets its conjugation substrates to the degradation by the 26S proteasome. This process termed FAT10ylation is mediated by an enzymatic cascade and includes the E1 activating enzyme ubiquitin-like modifier activating enzyme 6 (UBA6), the E2 conjugating enzyme UBA6-specific E2 enzyme 1 (USE1) and E3 ligases, such as Parkin. In this study, the function of the HECT-type ubiquitin E3 ligase HUWE1 was investigated as a putative E3 ligase and/or conjugation substrate of FAT10. Our data provide strong evidence that HUWE1 is FAT10ylated in a UBA6 and FAT10 diglycine-dependent manner *in vitro* and *in cellulo* and that the HUWE1-FAT10 conjugate is targeted to proteasomal degradation. Since the mutation of all relevant cysteine residues within the HUWE1 HECT domain did not abolish FAT10 conjugation, a role of HUWE1 as E3 ligase for FAT10ylation is rather unlikely. Moreover, we have identified the autophagy-related protein AMBRA1 as a new FAT10 interaction partner. We show that the HUWE1-FAT10 conjugate formation is diminished in presence of AMBRA1, while the interaction between AMBRA1 and HUWE1 is strengthened in presence of FAT10. This implies a putative interplay of all three proteins in cellular processes such as mitophagy.

## Introduction

The covalent modification of proteins with ubiquitin or ubiquitin-like (UBL) modifiers is an important post-translational modification which regulates virtually all cellular mechanisms in eukaryotic cells. Human leukocyte antigen (HLA)-F adjacent transcript 10 (FAT10) belongs to the family of ubiquitin-like modifiers whose members share the same three-dimensional structure, the so called β-grasp fold [1]. FAT10 contains two tandem arranged UBL domains at its N- and C terminus and has a molecular weight of 18.3 kDa. The UBL domains of FAT10 share 29% and 36% sequence identity with ubiquitin on amino acid level, respectively, and are connected by a short flexible linker [2, 3]. However, unlike ubiquitin, which is ubiquitously expressed, FAT10 has been detected in mammals only, where its expression is mainly

**Funding:** AA, SM: DFG Collaborative Research Center (SFB) 969 TP C01 and C09 https://www.dfg. de/ The funders had no role in study design, data collection and analysis, decision to publish, or preparation of the manuscript.

**Competing interests:** The authors have declared that no competing interests exist.

restricted to tissues and cells of the immune system such as thymus, lymph nodes or mature dendritic cells and B cells [3–6]. Moreover, FAT10 expression is synergistically induced in various tissues upon stimulation with the pro-inflammatory cytokines tumor necrosis factor (TNF) and interferon (IFN)-γ [7]. FAT10 has been described to play a role in antigen presentation, cell cycle control, NF-κB signaling, thymic T cell selection and additionally, it has been shown to possess pro-apoptotic as well as anti-apoptotic functions [8–16]. With its free diglycine motif at the C terminus FAT10 can directly bind to and covalently modify its target proteins for subsequent proteasomal degradation [17–19]. Similar to ubiquitylation, FAT10ylation is also regulated by a conjugation cascade involving the action of three enzymes. First, the E1 activating enzyme ubiquitin-like modifier activating enzyme 6 (UBA6) adenylates FAT10 at its C- terminal carboxyl group in an ATP dependent manner resulting in a thioester intermediate bound to the active site cysteine of the E1. The E2 conjugating enzyme UBA6-specific E2 enzyme 1 (USE1) then performs a transthiolation reaction by which FAT10 becomes transferred to the active site cysteine of USE1. Finally, E3 ligases, including Parkin, catalyze the covalent isopeptide bond formation with an internal lysine residue of a substrate [4, 20–25]. The ubiquitin E3 ligase HECT, UBA and WWE domain containing E3 ubiquitin protein ligase 1 (HUWE1) (also known as URE-B1, LASU1, ARF-BP1, Mule, or HectH9) [26–30] is a large protein of 4374 amino acids and has a molecular weight of 482 kDa. It is ubiquitously expressed and highly conserved in mammals [29]. At its N-terminus HUWE1 bears two domains of unknown function (DUF908 and DUF913) followed by a ubiquitin-associated (UBA) domain for the regulation of ubiquitin dependent proteolysis, WWE motifs to mediate specific protein-protein interactions and a highly conserved BH3 domain for interactions with pro- and anti-apoptotic members of the BCL-2 protein family [27, 31–33]. Within its C-terminal region HUWE1 harbors the characteristic "homologous to the E6-AP carboxyl terminus" (HECT) domain of about 350 amino acids that has a bilobed structure. Its amino-terminal N-lobe interacts with the charged E2 while the carboxy-terminal C-lobe contains the catalytic cysteine (Cys4341) [34–36]. HUWE1 catalyzes the ubiquitylation of diverse substrates thereby regulating the stability of a variety of cellular proteins and, as a consequence, it is involved in numerous physiological processes such as cell proliferation and differentiation, DNA replication and cell cycle progression, DNA damage response and repair [37–42]. HUWE1 as well as FAT10 are both highly upregulated in several cancer types [43–45]. Moreover, HUWE1 is activated upon stimulation with TNF, which in combination with IFN-γ also induces FAT10 expression [46]. Among the various HUWE1 ubiquitylation substrates there are some which are FAT10ylated or described as FAT10 interaction partners such as p53, Jun, Mfn2, PCNA and β-catenin [25, 47–51]. The activating molecule in beclin-1-regulated autophagy protein 1 (AMBRA1) was identified as interaction partner, but not ubiquitylation substrate, of the FAT10 E3 ligase Parkin [52]. AMBRA1 plays a role in both, canonical PINK1/Parkin-dependent and -independent mitophagy [53]. In a PINK1/Parkin-free context, AMBRA1 mediates the recruitment of HUWE1 to mitochondria, promotes binding of HUWE1 to its ubiquitylation substrate mitofusin-2 (Mfn2) and thereby supports mitophagy [54]. Mfn2 was also demonstrated to be FAT10ylated in a Parkin-dependent manner in SH-SY5Y cells, and it was suggested that HUWE1 might substitute for Parkin in Parkin-deficient cells such as HEK293 cells [25]. Based on these findings, we decided to investigate HUWE1 as a putative E3 ligase for FAT10 conjugation. We provide strong evidence that FAT10 covalently modifies HUWE1 and that this targets HUWE1 for proteasomal degradation. HUWE1 FAT10ylation is dependent on the FAT10 C-terminal diglycine motif, but independent of the active site cysteine of HUWE1, making an E3 ligase function of HUWE1 for FAT10ylation rather unlikely. Moreover, we have identified AMBRA1 as new non-covalent interaction partner of FAT10. We provide data showing that FAT10 strengthens the interaction between AMBRA1 and HUWE1

and that AMBRA1 negatively influences HUWE1-FAT10ylation, implying a putative interplay of the three proteins.

## Results

### FAT10 interacts in a diglycine-dependent manner with HUWE1

Our recent data have shown that FAT10 plays a role in mitophagy by using Parkin as E3 ligase and by inhibiting its ubiquitin E3 ligase function [25]. Since also HUWE1 was shown to be involved in mitophagy [54] and since HUWE1 and FAT10 share several common interaction partners, we speculated that there might be a connection between HUWE1 and FAT10, as well. To investigate this hypothesis, 6His-3xFLAG-tagged FAT10 (named hereafter as FLAG--FAT10) and a HA-tagged catalytically active and N-terminally truncated variant of HUWE1 (HUWE1ΔN, amino acid residues 2474–4374, Fig 1A) were transiently expressed in HEK293 cells. A subsequent co-immunoprecipitation confirmed an interaction between FAT10 and HUWE1ΔN under non-reducing and reducing (4% β-ME) conditions (Fig 1B top panel, IP: HA, IB: FLAG, lanes 4–5, 9–10). Reducing conditions diminished the intensity of the HUWE1-FAT10 conjugate while increasing the signal for monomeric FAT10 (Fig 1B, top and second panel, IP: HA, IB: FLAG, lanes 9–10) suggesting a thioester linkage, sensitive to reduction. Moreover, also a non-covalent interaction of FAT10 and HUWE1 was detected, both, under non-reducing and reducing (4% β-ME) conditions (Fig 1B, second panel, IP: HA, IB: FLAG, lanes 4–5, and 9–10). The addition of tumor necrosis factor (TNF), which was described to stimulate HUWE1 activity [46], slightly diminished conjugate formation in this specific experiment, while in other experiments this was not observed (Fig 1B, top panels, IP: HA, IB: FLAG lane 4, 5 or 9, 10). The conjugation of FAT10 to HUWE1 was only slightly detectable under endogenous conditions upon treatment with the pro-inflammatory cytokines TNF and IFN-γ (S1A Fig in S1 File).

In a next step the nature of this linkage was further specified and tested whether the active site cysteine of HUWE1 was involved in the interaction. Therefore, cells were transfected with plasmids expressing FLAG-FAT10 or its conjugation incompetent mutant FLAG-FAT10-AV, in which the two C-terminal glycine residues were mutated to alanine and valine, together with HA-tagged HUWE1ΔN, HUWE1ΔN-C/S or HUWE1ΔN-C/A in which the active site cysteine C4341 was mutated to serine (C/S) or alanine (C/A), respectively. As endogenous HUWE1 might distort the results by possibly forming dimers with overexpressed HUWE1 variants, this experiment was performed in newly generated HEK293 HUWE1-deficient (HUWE1-KO) cells (S1B Fig in S1 File). Immunoprecipitation and subsequent Western blot analysis showed that the conjugate formation was dependent on the diglycine motif of FAT10, as it was not detectable when the conjugation deficient FAT10-AV mutant was co-expressed (Fig 1C, top panel, IP: HA, IB: FLAG, lanes 7–8 and 11–12). However, a conjugate was also detected between FAT10 and both HUWE1 active site cysteine mutants under non-reducing and reducing (4% β-ME) conditions (Fig 1C, top panel, IP: HA, IB: FLAG lanes 9–10 and 13–14). Under the assumption that FAT10 would bind by a thioester to the active site cysteine of HUWE1, HUWE1ΔN-C/S should be capable of FAT10 binding but insensitive to reduction, while the C/A mutation should completely abolish the formation of this linkage. As FAT10 conjugation to both mutants was still detectable, we suggested that HUWE1 is rather a conjugation substrate than an E3 ligase for FAT10ylation.

To further investigate if FAT10 and HUWE1 interact directly with each other it was of interest to confirm HUWE1-FAT10ylation under *in vitro* conditions with recombinant proteins. HUWE1-KO cells were used to purify transiently expressed HA-HUWE1ΔN or HA-HUWE1ΔN-C/A by immunoprecipitation and the proteins were applied to a *semi-in vitro*

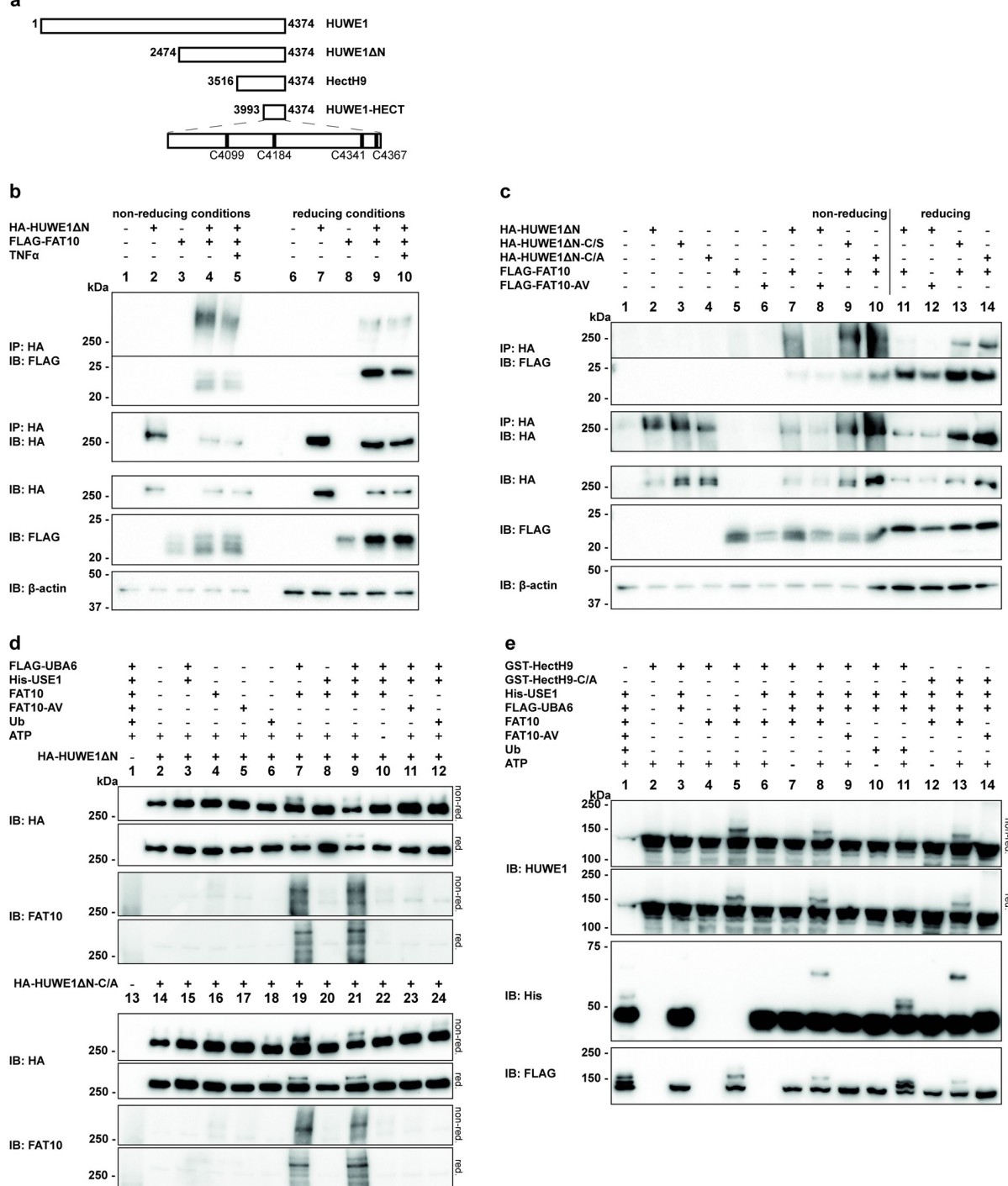

**Fig 1. FAT10 interacts covalently and non-covalently with HUWE1.** (a) Schematic presentation of the different HUWE1 constructs used in this study. The scheme at the bottom represents a magnification of the HUWE1-HECT domain to show the localization of the surface cysteine residues within this fragment. Cysteine 4341 represents the active site cysteine (b) HEK293 cells were transfected with expression plasmids for HA-HUWE1ΔN (amino acid residues 2474–4374) and FLAG-FAT10, and were additionally treated with TNF for 24 hours, as indicated. Cleared lysates were subjected to immunoprecipitation using a HA-reactive, agarose-coupled antibody. Samples were analyzed under non-reducing and reducing (4% β-ME) conditions by SDS-PAGE. For this purpose, immunoprecipitated samples were first boiled in 5x SDS PAGE sample buffer without β-ME, and then split into two portions. While one portion was left untreated (non-reducing conditions), 4% β-ME (final concentration) was added to the second portion and boiled again (reducing conditions). Subsequent Western blot analysis was performed with antibodies directed against the FLAG-tag or HA-tag, while β-actin was used as loading control. One representative experiment out of five experiments with similar outcomes is shown. (c) Transient protein expression of HA-tagged HUWE1ΔN variants and

FLAG-tagged FAT10 variants was induced in HEK293 HUWE1-KO cells by transfection. Immunoprecipitated samples were analyzed under non-reducing and reducing conditions by Western blot analysis using the antibodies indicated. One representative experiment out of four experiments with similar outcomes is shown. (d) HA-tagged HUWE1ΔN variants were immuno-precipitated using anti-HA agarose from cleared lysates of HEK293 HUWE1-KO cells which were transfected with the corresponding expression plasmids. The proteins were used for a *semi-in vitro* FAT10ylation assay together with recombinant FLAG-UBA6, 6His-USE1, tagless FAT10 variants and ubiquitin. Conjugate formation was visualized by Western blot analysis using anti-HA and polyclonal anti-FAT10 antibodies. One representative experiment out of two replicates with similar outcomes is shown. (e) For *in vitro* FAT10ylation assays recombinant GST-tagged HectH9 variants (amino acid residues 3516–4374, Fig 1A) were incubated with recombinant 6His-USE1, FLAG-UBA6, tagless FAT10 variants or ubiquitin for 30 min at 37˚C. Subsequent SDS-PAGE was followed by Western blot analysis using anti-FLAG, anti-His and anti-HUWE1 (ARF-BP1) reactive antibodies. Shown is one representative experiment out of three with similar outcomes.

FAT10ylation assay together with recombinant FLAG-UBA6, His-USE1 and tagless FAT10 or FAT10-AV. The formed conjugate was displayed under non-reducing and reducing conditions by Western blot analysis. A slightly reducible conjugate of FAT10, but not FAT10-AV, with both, HUWE1ΔN and the HUWE1ΔN-C/A mutant was observed in an ATP-dependent manner (Fig 1D, IB: FAT10 and IB: HA). Of note, recombinant UBA6 alone was sufficient to initiate the conjugate formation between FAT10 and HUWE1 (Fig 1D, IB: HA, lanes 7 and 19). The addition of USE1 did not further increase the amount of FAT10ylated HUWE1 (Fig 1D, IB: HA, lanes 9 and 21), as observed already previously for other FAT10 substrates [55–58]. A transfer of ubiquitin onto HUWE1 was not detectable under these conditions (Fig 1D, IB: HA, lanes 12 and 24).

These results were further confirmed in an *in vitro* FAT10ylation assay with recombinant GST-tagged N-terminally truncated HUWE1 (HectH9, amino acid residues 3516–4374, Fig 1A), and its active site cysteine mutant HectH9-C/A. Using these proteins, the *in vitro* conjugate formation was likewise detected with both HUWE1 variants in presence of FAT10, but not in presence of FAT10-AV (Fig 1E, IB: HUWE1, lanes 5,8,9,13,14). A transfer of ubiquitin was again not detectable under these conditions (Fig 1E, IB: HUWE1, lane 11). In summary, our data revealed that HUWE1 is covalently modified with FAT10 under *in cellulo* and *in vitro* conditions, dependent on the C-terminal diglycine motif of FAT10. Interestingly, the conjugate turned out to be at least partly sensitive to reduction, although the modification was not dependent on the active site cysteine of HUWE1, which is described to be indispensable for ubiquitin transfer. However, due to the observed sensitivity to β-ME, we do not want to completely rule out a putative E3 ligase function for FAT10 conjugation.

## Bulk FAT10 conjugation is not affected by HUWE1

To further investigate a putative E3 ligase function of HUWE1 for FAT10ylation, HEK293 cells were transiently transfected with expression plasmids for FLAG-tagged FAT10 or FLAG-FAT10-AV together with HA-tagged HUWE1ΔN or its active site cysteine (C/A) mutant. FAT10-conjugates were immunoprecipitated with a FLAG-reactive antibody and visualized by Western blot analysis (Fig 2A). The amount of FAT10 conjugates was additionally quantified by densitometric analysis of the Western blot ECL signals (Fig 2B). When HUWE1 was over-expressed, no impact on the total FAT10 conjugate amount was visible. Also, the band pattern of detected FAT10 conjugates did not visibly change (Fig 2A, lanes 4 vs 6), suggesting that HUWE1 might either not act as a major E3 ligase for FAT10 conjugation or that the amount of HUWE1 was not limiting in HEK293 cells, so that the additional overexpression did not further affect FAT10 conjugation. Moreover, the knockout of HUWE1 did not significantly diminish FAT10 conjugation, although these cells showed a tendency to form slightly decreased amounts of overall FAT10 conjugates when compared to wild type cells (Fig 2A, upper panel, IP: FLAG, IB: FLAG, lane 4 vs 10, and Fig 2B). However, FAT10 conjugation did not increase in HUWE1-KO cells reconstituted with a HUWE1 expression plasmid (Fig 2A,

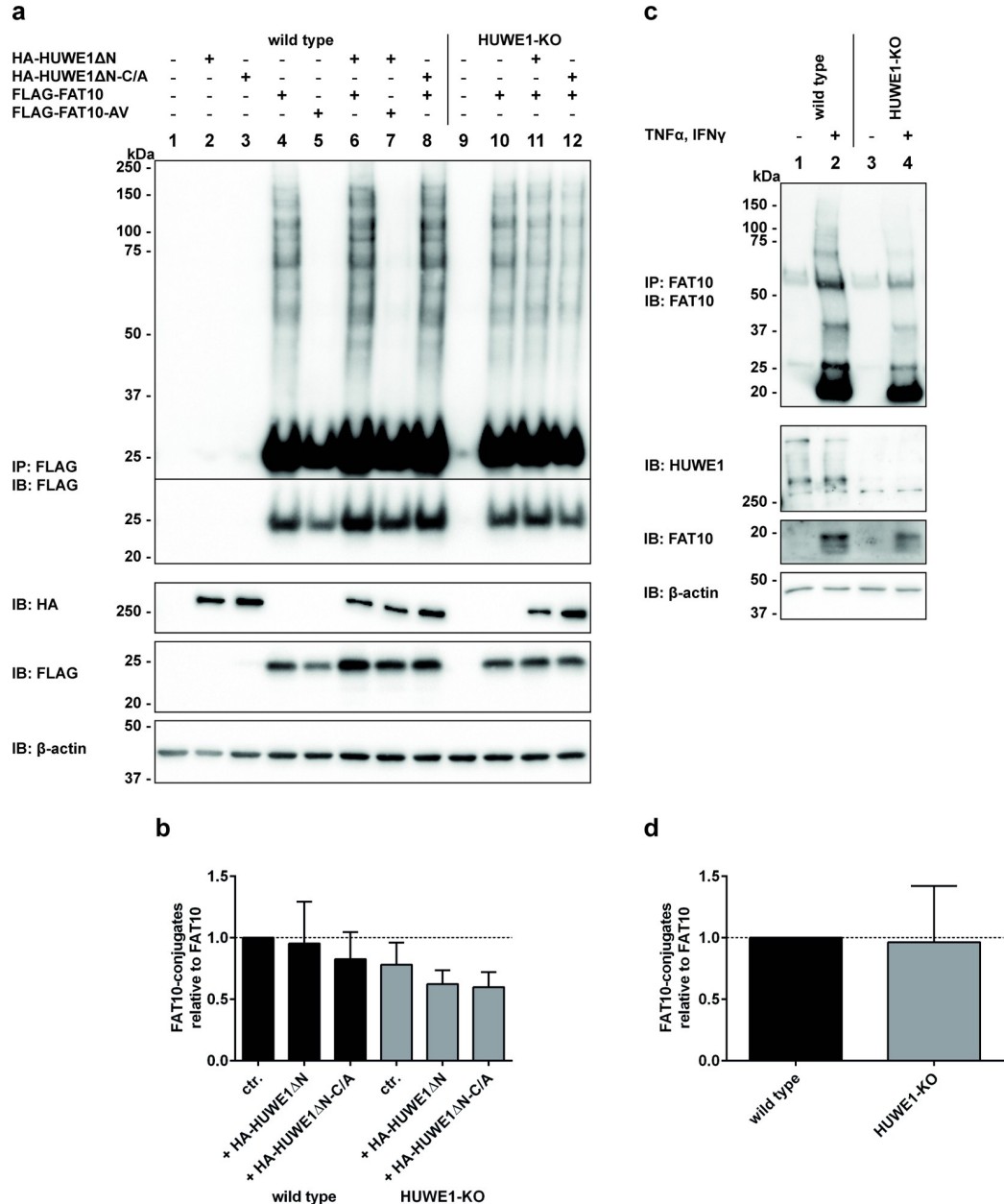

**Fig 2. HUWE1 has no impact on bulk FAT10 conjugates.** (a) Both HEK293 wild type and HEK293 HUWE1-KO cells were transfected with expression plasmids for HA-HUWE1ΔN variants and FLAG-FAT10 variants. Immunoprecipitation from cleared lysates was performed using Anti-FLAG M2 Affinity Gel and proteins were analyzed by SDS-PAGE and Western blotting with antibodies directed against the FLAG-tag and the HA-tag. β-actin was used as loading control. One representative experiment out of three replicates with similar outcomes is shown. (b) Densitometric analysis of the ECL signals shown in (a). The graph shows the amount of total FAT10 conjugates normalized to the amount of monomeric FAT10. Levels of untreated wild type cells expressing FAT10 were set to 1 and data from at least three independent experiments were analyzed for each condition. (c) Endogenous expression of FAT10 was induced in HEK293 wild type and HUWE1-KO cells by treatment with TNF and IFN-γ for 24 hours. FAT10 conjugates were immunoprecipitated with a monoclonal anti-FAT10 (4F1) antibody coupled to protein A sepharose and samples were subjected to SDS-PAGE followed by Western blotting. Displayed is one representative experiment out of three with similar outcomes. (d) The graph shows the quantification of the ECL signals of the Western blot signals shown in (c). Densitometry analysis was performed with values for three independent experiments.

IP: FLAG, IB: FLAG, lane 10 vs 11, and Fig 2B). As a control, cells were transfected with the diglycine mutant of FAT10 (FAT10-AV) and as expected, no FAT10 conjugation was detected (Fig 2A, IP: FLAG, IB: FLAG, lanes 5, 7). Unexpectedly, overexpression of the active site cysteine mutant of HUWE1 (HA-HUWE1ΔN-C/A) also led to a slight, however not significant reduction of FAT10 conjugates in both, HEK293 wild type and HUWE1-knockout cells (Fig 2A, IP: FLAG, IB: FLAG, lane 8 and 12, and Fig 2B). To further investigate the effect of HUWE1 on FAT10 conjugation, the expression of endogenous FAT10 in both, HEK293 and HUWE1-KO cells was induced by cell treatment with TNF and IFN-γ. Also under these endogenous conditions, no difference in the FAT10 conjugate amount between wild type and HUWE1-knockout cells was observed (Fig 2C and 2D). As a last approach, cells were treated with the previously published small molecule inhibitors BI8622 and BI8626 [59] to block endogenous HUWE1 activity. However, also this did not generate a visible effect on overall FAT10 conjugation (S2 Fig in S1 File).

Taken together, no impact of overexpressed HUWE1 was observed on bulk FAT10 conjugate formation and in addition, FAT10 was still conjugated to substrates in the absence of functional HUWE1. Therefore, we suggest that HUWE1 might not be a major E3 ligase for bulk FAT10 conjugation in HEK293 cells but maybe act in a cell type specific manner or might be specific for a certain subset of particular substrates.

## FAT10 transfer to HUWE1 is independent of HECT domain surface cysteines

Mutation of the active site cysteine C4341 of HUWE1 to either alanine or serine had no influence on the conjugate formation between HUWE1 and FAT10 (Fig 1), although this cysteine was described to be the specific active site for ubiquitin conjugation [28, 29]. Nevertheless, we always observed lower HUWE1-FAT10 conjugate amounts under reducing conditions (4% β-ME). Thus, we were wondering if in the HUWE1 HECT domain another cysteine besides C4341 might have an activity towards FAT10 conjugation. To consider this possibility, several cysteine mutants of HUWE1 were created by site-directed mutagenesis using an expression construct that is mainly restricted to the catalytically active HECT domain (amino acid residues 3993–4374). Six cysteine residues are found in the HECT domain, while four of them are clearly located on the surface (Cys4099, Cys4184, Cys4341, Cys4367), Cys4211 is located inside and another cysteine (Cys4126) might have little access to the surface (PDB ID: 5LP8 [60]). Recombinant 6His-tagged HUWE1-HECT (His-HUWE1-HECT), its active site cysteine mutant (His-HUWE1-HECT C/A, Cys4341), a triple mutant (His-HUWE1-HECT TM), in which the three cysteine residues (Cys4099, 4184, 4367) located on the surface of the HUWE1 HECT domain were mutated to alanine residues [36], as well as a quadruple mutant with additionally mutated active site cysteine in the TM mutant (His-HUWE1-HECT QM), were subjected to *in vitro* FAT10ylation assays together with recombinant FLAG-UBA6, 6xHis-USE1 (His-USE1) and FAT10 (Fig 3A). Again, a partially reducible conjugate of FAT10 and His-HUWE1-HECT as well as His-HUWE1-HECT-C/A mutant was formed (Fig 3A, lanes 3–4 and 6–7). However, FAT10 conjugates were also detected with His-HUWE1-HECT TM as well as -QM, showing comparable ECL signals in case of all formed conjugates (Fig 3A, lanes 9–10 and 12–13). As a control, the active site cysteine mutant C4341A did not accept ubiquitin from its cognate E1 UBE1 and E2 UbcH5b (S3A Fig in S1 File, lanes 2–4) verifying the correct folding of the recombinant HECT domain proteins. Interestingly, UBA6 and USE1, which are both bispecific for FAT10 and ubiquitin [20, 21, 23], were capable of transferring a single ubiquitin onto HUWE1-HECT and also onto the active site cysteine mutant (S3A Fig in S1 File, lanes 6–8), pointing to a possible covalent ubiquitylation of HUWE1 in presence of UBA6 and

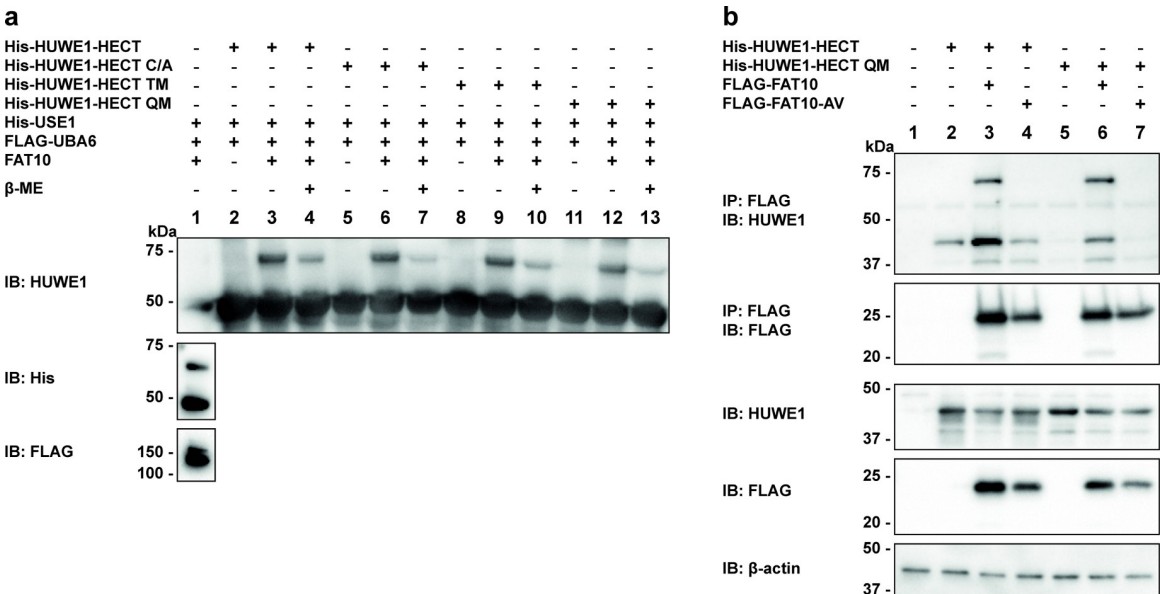

**Fig 3. The interaction between HUWE1 and FAT10 is independent of HECT surface cysteines.** (a) *In vitro* FAT10ylation assays were performed with recombinant cysteine mutants of 6His-HUWE1-HECT (amino acid residues 3993–4374). Recombinant His-tagged HUWE1-HECT wild type or its cysteine mutants C/A, TM or QM, was incubated with recombinant 6His-USE1, FLAG-UBA6 and tagless FAT10 at 37°C for 30 min. Samples were split and analyzed under non-reducing and reducing (4% β-ME) conditions on SDS-PAGE followed by Western blot analysis with anti-HUWE1 (ARF-BP1), anti-His and anti-FLAG-reactive antibodies. Shown is one representative experiment out of four with similar outcomes. (b) His-tagged HUWE1-HECT wild type or quadruple mutant (QM) in combination with FLAG-tagged FAT10 or its AV mutant were transiently expressed in HEK293 HUWE1-KO cells. After 24 hours, co-immunoprecipitation was performed from cleared lysates using Anti-FLAG M2 Affinity Gel. Samples were subjected to SDS-PAGE and subsequent Western blotting with the corresponding antibodies reactive against HUWE1 (ARF-BP1) and FLAG as well as anti-β-actin as loading control. Shown is one representative experiment out of two with similar outcomes.

USE1. Of note, to our knowledge this is the first time that a transfer of UBA6-activated ubiquitin onto HUWE1 was shown.

The conjugate formation between FAT10 and the above described His-HUWE1-HECT variants was further confirmed in HEK293 HUWE1-KO cells transiently expressing FLAG-FAT10 variants and His-HUWE1-HECT wild type or -QM, respectively (Fig 3B and S3B Fig in S1 File). Again, the formation of the conjugate required the C-terminal glycine of FAT10, as it was only detectable in presence of wild type FAT10 and HUWE1-HECT wild type and -QM, but absent when FLAG-FAT10 AV was expressed instead (Fig 3B, upper panel, IP: FLAG, IB: HUWE1, lanes 3, 6 vs 4, 7). Furthermore, as already shown under *in vitro* conditions (Fig 3A), formation of the HUWE1-FAT10 conjugate was confirmed for all HUWE1-HECT variants in *in cellulo* experiments (S3B Fig in S1 File).

In summary, our data confirmed that the transfer of FAT10 onto HUWE1 is independent of all tested surface cysteines in the HUWE1 HECT domain. Since nevertheless the conjugate appeared to be partially reducible in presence of 4% β-ME, we cannot exclude that FAT10 was eventually transferred onto cysteine residue 4126, which might be partially accessible at the surface of the HECT domain.

## FAT10ylation of HUWE1 targets it to degradation by the 26S proteasome

As described in previous publications, FAT10ylation of most proteins causes their degradation via the 26S proteasome [18, 24]. For this reason, it was examined whether this also holds true for the HUWE1-FAT10 conjugate. To monitor proteasomal degradation of FAT10ylated

HUWE1, a cycloheximide (CHX) chase assay was performed with HEK293 HUWE1-KO cells expressing His-HUWE1-HECT or the -QM mutant together with FLAG-FAT10 or FLAG-FAT10-AV. As a control, where indicated in Fig 4A, the proteasome inhibitor MG132 was added to block the catalytic activity of the 26S proteasome. Confirming results from earlier publications [18, 20], unconjugated FAT10 was degraded over time and its degradation was prevented when the cells were treated at the same time with the proteasome inhibitor MG132 (Fig 4A, IP: FLAG, IB: FLAG). Similarly, HUWE1-FAT10 conjugates formed with HUWE1-HECT or HUWE1-HECT QM, were degraded over time and their degradation was likewise rescued by MG132 treatment (Fig 4A, upper panel, IP: FLAG, IB: HUWE1, lanes 5–8 and 10–13). Densitometric analysis of the Western blot ECL signals further confirmed our results of FAT10ylated HUWE1 being degraded by the 26S proteasome (Fig 4B upper panel), while unmodified HUWE1-HECT and HUWE1-HECT QM were stable and not degraded in the presence or absence of FAT10 (Fig 4B, lower panel and Fig 4C).

## The HUWE1- and mitophagy-related protein AMBRA1 is a new interaction partner of FAT10 and negatively interferes with HUWE1-FAT10ylation

Previous studies have not only shown an interaction of the autophagy regulator AMBRA1 with Parkin, but also with HUWE1 [52, 54]. Since both proteins, HUWE1 and Parkin [25] were also shown to interact with FAT10, we were wondering whether AMBRA1 could also be an interaction partner of FAT10. To investigate the interaction between FAT10 and AMBRA1 under *in cellulo* conditions, FLAG-tagged AMBRA1 was expressed together with HA-FAT10 in HEK293 cells followed by co-immunoprecipitation and subsequent Western blot analysis. As shown in Fig 5A, AMBRA1 was co-immunoprecipitated with FAT10 and the interaction of AMBRA1 with FAT10 was independent of the FAT10 diglycine motif (Fig 5A, IP: HA, IB: FLAG, lanes 4, 6). Interestingly, AMBRA1 was not covalently modified with FAT10, but interacted non-covalently with FAT10 (Fig 5A and 5B, lane 6). The same results were obtained when ubiquitin instead of FAT10 was expressed and co-immunoprecipitated together with AMBRA1 (Fig 5B, IP: HA, IB: FLAG, lane 4). A reverse immunoprecipitation confirmed these results (Fig 5C).

Following these observations, we investigated if the presence of FAT10 might have an impact on the interaction between AMBRA1 and HUWE1. For this purpose, AMBRA1-FLAG was expressed in HEK293 HUWE1-KO cells together with either His-HUWE1-HECT or His-HUWE1ΔN in combination with tagless FAT10 or the FAT10-AV mutant. Cell lysates were subjected to co-immunoprecipitation followed by Western blot analysis (Fig 6A and 6B). A non-covalent interaction of AMBRA1 with HUWE1 was observed for both HUWE1 variants (Fig 6A and 6B, IP: FLAG IB: HUWE1, lanes 6–8, each), confirming the earlier finding by Di Rita and colleagues [54]. Interestingly, the quantification of the ECL signals obtained for both, HUWE1-HECT as well as HUWE1ΔN, revealed that the interaction between AMBRA1 and HUWE1 was enhanced in the presence of FAT10 and FAT10-AV, albeit not significantly and to a lesser extent with FAT10-AV (Fig 6C). The tendency towards a stronger FAT10-dependent interaction between AMBRA1 and HUWE1 was further verified under endogenous conditions in HEK293 wild type cells. Upon cell treatment with TNF and IFN-γ, to induce expression of endogenous FAT10, the amount of endogenous HUWE1 co-immunoprecipitated with FLAG-tagged AMBRA1 was increased, as well (Fig 6D and 6E). As we had clearly identified FAT10 as interaction partner of both HUWE1 and AMBRA1, we further investigated if AMBRA1 might influence HUWE1-FAT10ylation. Therefore, the formation of the covalent HUWE1-FAT10 conjugate was examined in the absence or presence of AMBRA1. As

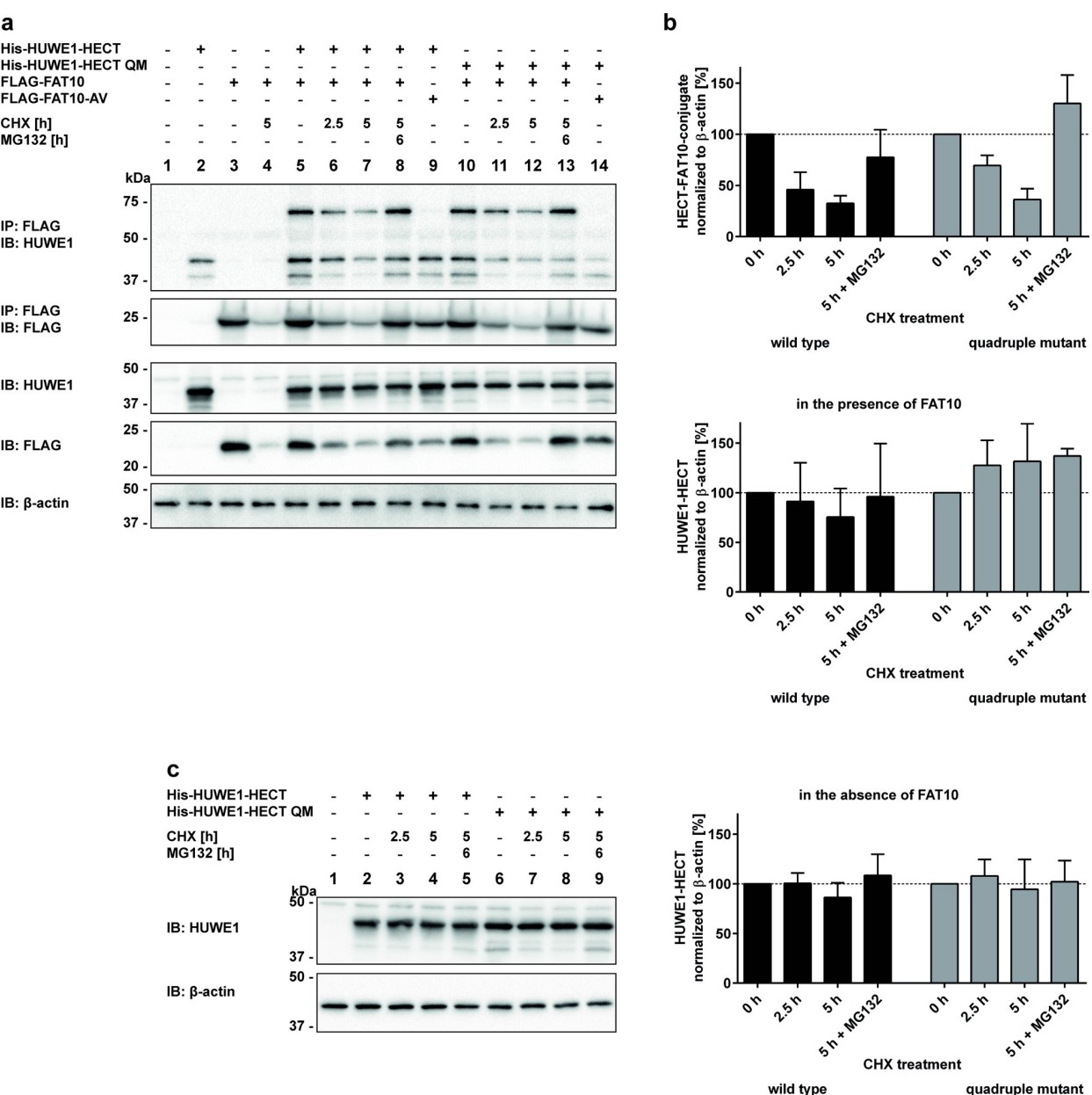

**Fig 4. FAT10ylated HUWE1 is degraded by the 26S proteasome.** (a) HEK293 HUWE1-KO cells transiently expressing His-HUWE1-HECT wild type or quadruple mutant (QM) and FLAG-FAT10 or -AV were treated for the indicated times with cycloheximide (CHX) and/or the proteasome inhibitor MG132. Cleared lysates were subjected to immunoprecipitation using Anti-FLAG M2 Affinity Gel and proteins were separated by SDS-PAGE. Western blot analysis was performed with anti-HUWE1 (ARF-BP1) and anti-FLAG antibodies, and β-actin was used as loading control. (b) For densitometry determinations, Western blot ECL signals were quantified, and the graph shows the amounts of HUWE1-FAT10 conjugates (upper panel), and the amount of HUWE1-HECT and–QM in the presence of FAT10 (lower panel), normalized to the respective amount of β-actin. The values of untreated cells were set to 100%. Shown is one representative experiment out of three replicates with similar outcomes. (c) HEK293 HUWE1-KO cells were transfected with expression constructs for His-HUWE1-HECT wild type or -QM, and treated for the indicated times with CHX and/or MG132. Proteins were detected by Western blot analysis with anti-HUWE1 (ARF-BP1) and anti-β-actin antibodies. The protein amounts from four replicates with similar outcomes were quantified by densitometry analysis as described in (b).

shown in Fig 6F, FAT10ylation of HUWE1-HECT was significantly decreased upon AMBRA1 co-expression (Fig 6F, IP: HA, IB: HUWE1, lane 5 vs 6 and quantification in Fig 6G). A comparable effect was observed, when HUWE1-HECT was exchanged by the larger His-

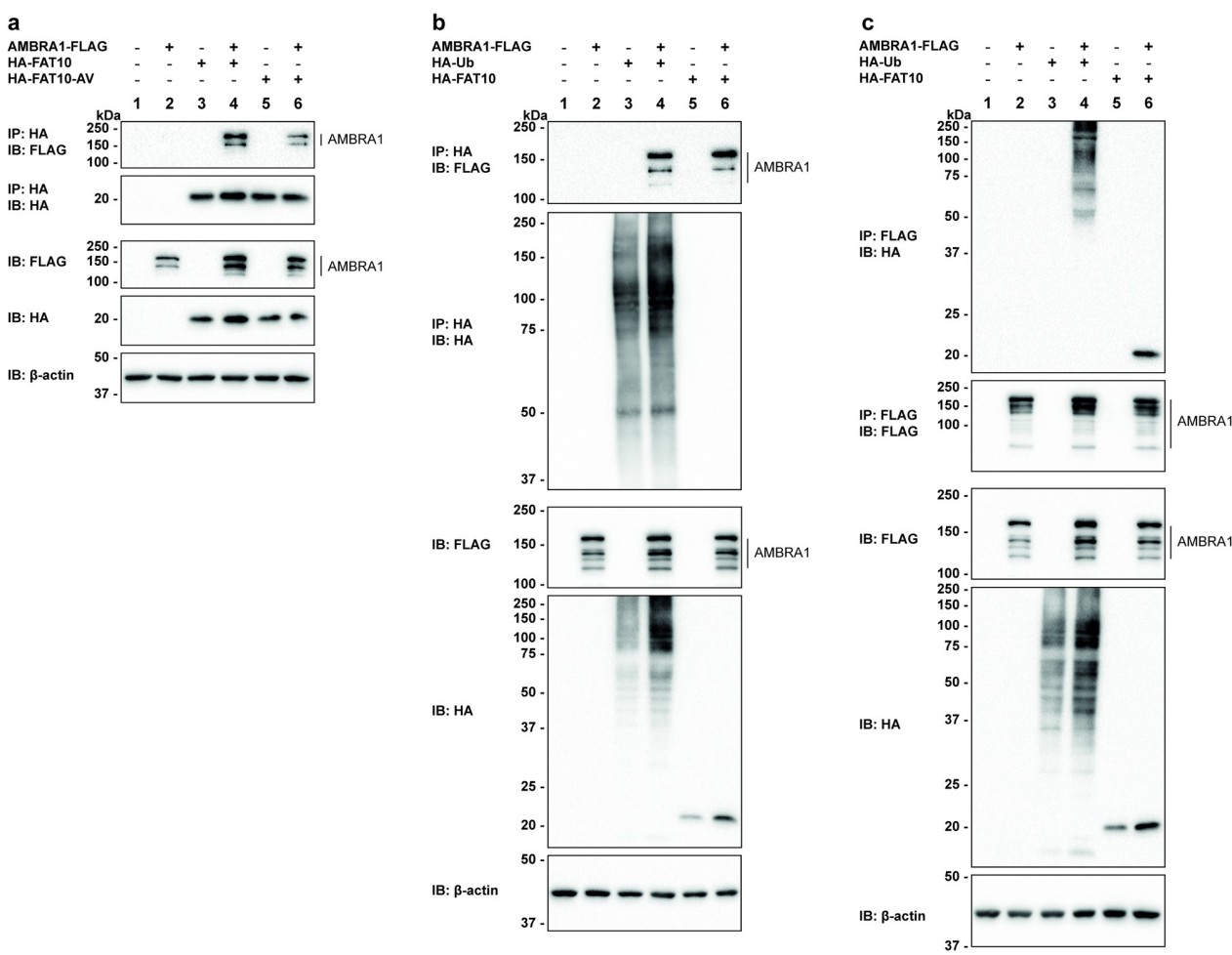

**Fig 5. AMBRA1 is a new FAT10 non-covalent interaction partner.** (a) HEK293 cells were transfected with expression plasmids for AMBRA1-FLAG and HA-FAT10 or HA-FAT10-AV. Cleared lysates were subjected to immunoprecipitation using HA-reactive, agarose-coupled anti-bodies. Subsequent Western blot analysis was performed with antibodies against the FLAG-tag or HA-tag while β-actin was used as loading control. One representative experiment out of three experiments with similar outcomes is shown. (b and c) To study the interaction between AMBRA1 with either ubiquitin or FAT10, FLAG-tagged AMBRA1 in combination with either HA-tagged ubiquitin or FAT10 were transiently expressed in HEK293 cells. Co-immunoprecipitation was performed with HA-reactive, agarose-coupled antibodies (b) or Anti-FLAG M2 Affinity Gel (c). Proteins were detected with antibodies reactive against the FLAG- or HA-tag, and β-actin was used as loading control. One representative experiment each out of three with similar outcomes is shown.

HUWE1ΔN construct (S4 Fig in S1 File), confirming that AMBRA1 negatively regulates HUWE1 FAT10ylation. In summary, these data provide strong evidence that the interaction between AMBRA1 and HUWE1 is stronger in presence of FAT10. Moreover, we have identified AMBRA1 as new non-covalent interaction partner of FAT10 and show that in the presence of AMBRA1 the HUWE1-FAT10 conjugate formation is diminished.

## Discussion

In the present study, we have identified the ubiquitin E3 ligase HUWE1 as a new conjugation substrate as well as non-covalent interaction partner of FAT10. Moreover, we confirm the autophagy- and mitophagy-related protein AMBRA1 as new non-covalent FAT10 interaction partner. We provide strong evidence that AMBRA1 negatively influences HUWE1 FAT10yla-tion and that in turn, FAT10 strengthens the AMBRA1-HUWE1 interaction, suggesting an

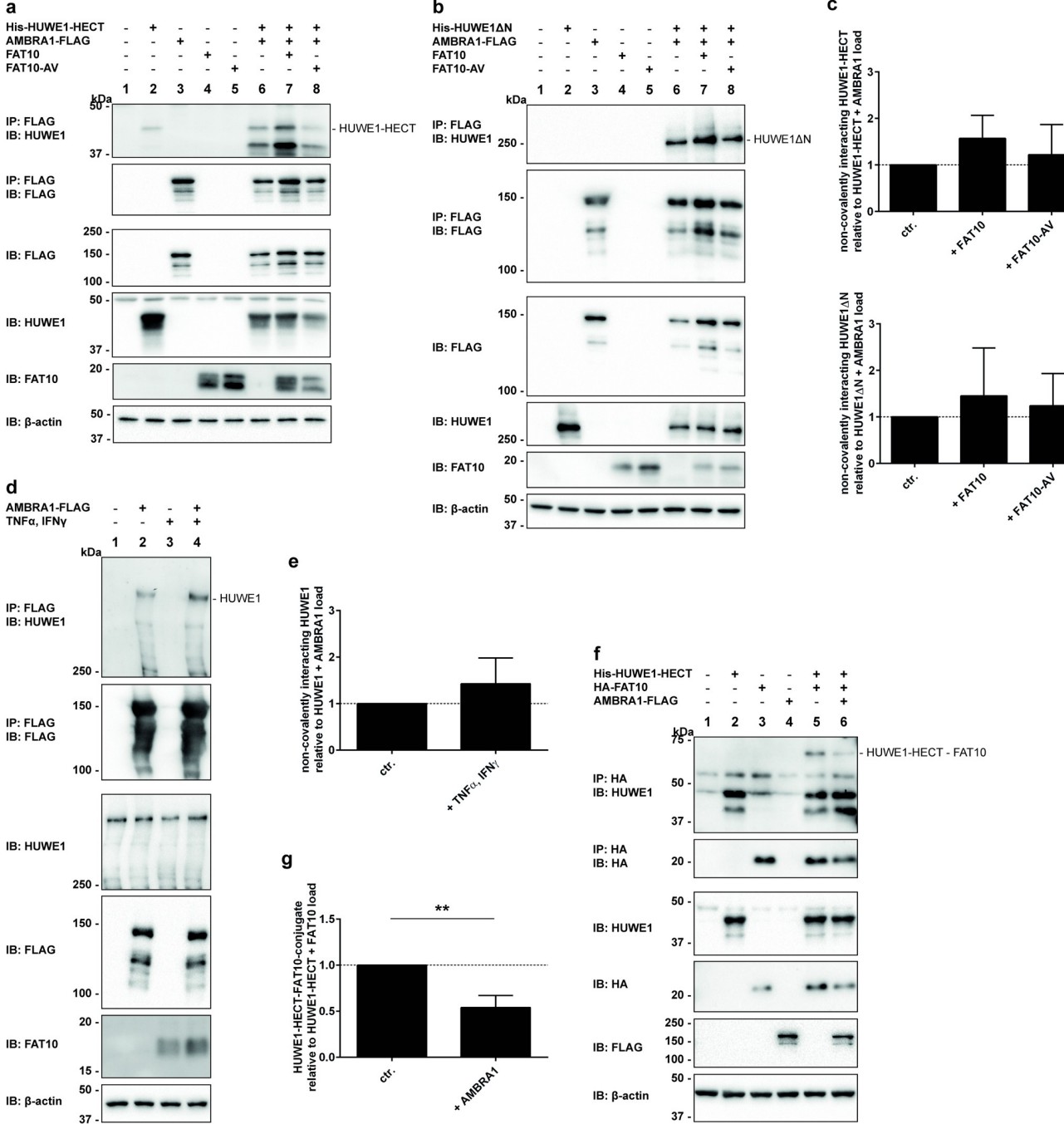

**Fig 6. FAT10 positively influences the HUWE1-AMBRA1 interaction.** (a) His-tagged HUWE1-HECT, FLAG-tagged AMBRA1 and FAT10 variants were expressed in HEK293 HUWE1-KO cells by transfection. Immunoprecipitated samples were analyzed by SDS-PAGE and Western blot analysis using the antibodies indicated. One representative experiment out of three experiments with comparable outcomes is shown. (b) The influence of FAT10 on the interaction between AMBRA1 and HUWE1 was assessed by transient protein expression of His-tagged HUWE1ΔN, FLAG-tagged AMBRA1 and FAT10 variants in HEK293 HUWE1-KO cells. Cleared lysates were subjected to immunoprecipitation followed by SDS-PAGE and Western blot analysis with antibodies against FLAG, HUWE1 (ARF-BP1), FAT10 and β-actin. Displayed is one representative experiment out of three with similar outcomes. (c) Quantification of (a) and (b). Densitometry analysis was performed to quantify the ECL signals from the Western blot. The graph shows the amount of non-covalently interacting HUWE1-HECT or HUWE1ΔN normalized to the amount of AMBRA1 and HUWE1 in the load. (d) Endogenous HUWE1 was immunoprecipitated with transiently expressed AMBRA1-FLAG while endogenous expression of FAT10 was induced by treatment with TNF and IFN-γ for 24 hours. Proteins were separated by SDS-PAGE and analyzed by western blotting with antibodies directed against HUWE1 (ARF-BP1), FLAG-tag and FAT10, as well as β-actin as loading control. Displayed is one representative experiment out of three with similar outcomes. (e) The graph shows the quantification from densitometry analysis of Western blot ECL signals displayed in (d) as described above. Densitometry analysis was performed with

values of three independent experiments. (f) Western blot showing co-immunoprecipitation experiments of His-HUWE1-HECT and HA-FAT10 in the presence of AMBRA1-FLAG from the cleared lysate of HEK293 HUWE1-KO cells which were transfected with the corresponding expression plasmids. Displayed is one representative experiment out of three with comparable outcomes. (g) Quantification from densitometry analysis of Western blot signals shown in (f) as described above. Densitometry analysis was per-formed with values for three independent experiments.

interplay of these three proteins. The initial hypothesis of HUWE1 being an E3 ligase for FAT10 conjugation could neither be confirmed nor completely be excluded since overexpression or CRISPR/Cas9-mediated knockout of HUWE1 in HEK293 cells had only a neglectable effect on bulk FAT10 conjugation (Fig 2). Admittedly, only minor effects of HUWE1 overexpression or deletion on overall FAT10 conjugation were expected because hundreds of E3 ligases have been discovered for ubiquitin [61] and presumably the same applies to FAT10, although up to date, only one single FAT10 E3 ligase could be confirmed, namely Parkin [25]. Another argument speaking against a putative E3 ligase activity for FAT10ylation was our observation that wild type HUWE1, as well as the catalytically inactive HUWE1 variant both accepted UBA6-activated FAT10 under *in vitro* and *in cellulo* conditions (Fig 1). Under reducing conditions in the presence of 4% β-ME, the HUWE1-FAT10 conjugate was at least partially reducible, initially suggesting the formation of a thioester linkage. To further clarify the nature of this linkage, HUWE1 mutants were generated, in which additional cysteine residues, which all are predicted to appear on the surface of the HUWE1 HECT domain, were mutated to alanine (PDB ID: 5LP8 [60]). The idea was that a cysteine nearby the active site cysteine might be able to accept FAT10, as it has been recently described for ubiquitylation with a RING-type ubiquitin E3 ligase. Within this so called RING-Cys-Relay mechanism ubiquitin was relayed from an upstream cysteine to a downstream cysteine by an intramolecular trans-thiolation reaction [62]. However, since FAT10 was still transferred onto the HUWE1 mutant lacking all four surface cysteine residues in the HECT do-main (quadruple mutant, HUWE1--HECT QM, Fig 3) this hypothesis seems rather unlikely. On the other hand, FAT10 might eventually act in a non-canonical way and be able to switch between cysteine residues in case the preferred cysteine is not accessible, as this was observed already for the isopeptide bond formation with lysines [63]. Since the active site cysteines are normally highly conserved, also this scenario seems rather unlikely. However, another explanation might be that FAT10 was transferred onto cysteine residue 4126, which might be partially accessible at the surface of the HECT domain (PDB ID: 5LP8 [60]). Interestingly, also other groups have observed that a mutation of the active site cysteine does not necessarily completely abolish ubiquitin transfer onto HUWE1 [30, 36]. We saw the same in our experiments, where we also detected a transfer of UBA6-activated ubiquitin onto the HUWE1 active site cysteine mutant (S3A Fig in S1 File). Taken these findings together and taking the slightly decreased FAT10 conjugate amount in HUWE1 knockout cells into account, we cannot completely rule out the possibility that HUWE1 might have E3 ligase activity for some specific, yet unidentified FAT10 conjugation substrates.

In a cycloheximide chase assay FAT10ylated HUWE1 was found to be degraded via the pro-teasomal pathway (Fig 4). As earlier shown for several other FAT10 conjugation substrates such as PDE6, OTUB1 or p62, degradation was inhibited when the proteasome was inhibited at the same time with the proteasome inhibitor MG132 [18, 56, 58, 64]. However, since only a small portion of HUWE1 becomes FAT10ylated and degraded (Fig 4), it is rather difficult to investigate putative functional consequences of FAT10-mediated HUWE1 degradation, especially because HUWE1 has dozens of different functions in cells.

Recent studies have shown that FAT10 can either positively or negatively influence the activity of its non-covalent interaction partners such as OTUB1 or PDE6 [56, 58]. Thus, we posed

the question if FAT10 might influence the activity of HUWE1, as well. HUWE1 and FAT10 were both recently shown to have specific functions in mitophagy [25, 54]. While FAT10 uses the E3 ligase Parkin for FAT10ylation of the mitochondrial protein mitofusin-2 (Mfn2), HUWE1 was shown to promote AMBRA1-mediated mitophagy by regulating the activity of AMBRA1 [54]. Interestingly, we have identified AMBRA1 as a new FAT10 non-covalent interaction partner (Fig 5). Since AMBRA1 was furthermore described to interact with both E3 ligases, Parkin and HUWE1 [52, 54], we decided to investigate a possible interplay of these three proteins. Indeed, in the presence of FAT10 a stronger interaction between AMBRA1 and HUWE1 was observed, while at the same time, the HUWE1-FAT10 conjugate formation was negatively influenced by AMBRA1, pointing to a mutual interaction of the three proteins. Interestingly, the HUWE1 HECT domain was sufficient as interaction site for the non-covalent binding of AMBRA1 to HUWE1 (Fig 6A). Additionally, the lysine residue used for FAT10ylation is most likely also located within the HECT domain of HUWE1 (Fig 3). Thus, upcoming experiments will show, if FAT10 might be able to interact with one of its UBL domains with AMBRA1, while maybe at the same time interacting with its second UBL domain with HUWE1, causing the formation of a trimeric complex. This idea is supported by earlier studies where it was shown that the FAT10 N-terminal UBL domain interacted with the three UBA domains of NEDD8-ultimate buster 1 long (Nub1L), while it was at the same time able to bind with its C-terminal UBL domain to the von Willebrand factor A (VWA) domain of the proteasome subunit Rpn10. Consequently, a trimeric complex of FAT10, Nub1L and the proteasome is formed, leading to a faster degradation of FAT10 and FAT10ylated proteins [65].

Upregulation of FAT10 expression was recently shown to inhibit the Parkin E3 ubiquitin ligase activity causing for example reduced ubiquitylation of mitofusin-2. This resulted in a diminished removal of damaged mitochondria and consequently in an increased neuronal cell death, pointing to a negative impact of FAT10 on PINK1/Parkin-mediated mitophagy [54]. Moreover, HUWE1 and AMBRA1 were described to play a role in PINK1/Parkin-independent mitophagy and HUWE1-mediated mitofusin-2 ubiquitylation was shown to be increased in presence of AMBRA1 [54]. Since our results showed a tendency towards a stronger interaction of HUWE1 and AMBRA1 in the presence of FAT10, one might expect that in PINK1/Parkin-independent mitophagy, FAT10 might rather have a positive effect on mitophagy. Speaking against this idea is the finding that the HUWE1-FAT10 conjugate was degraded by the proteasome, pointing to a very complex regulation of the different mitophagy pathways. Therefore, it will be interesting to address the question on the interplay of FAT10, HUWE1 and AMBRA1 in PINK1/Parkin-independent mitophagy in future experiments.

## Materials and methods

### Human cell culture

HEK293 (ATCC® CRL-1573™) and HEK293 CRISPR/Cas9 HUWE1-KO cells were cultured in Iscove's Modified Dulbecco's Medium (IMDM, PAN-Biotech) supplemented with 10% fetal calf serum (Gibco/Thermo Fisher Scientific), 1% stable glutamine (100x, 200 mM), and 1% penicillin/streptomycin (100x) (both from PAN-Biotech).

### Plasmids

The following constructs were used to transiently express proteins in HEK293 cells. For FLAG-tagged, HA-tagged or tagless FAT10: pcDNA3.1-His-3xFLAG-FAT10 [21], pcDNA3.1-His 3xFLAG FAT10 AV (C-terminal AV instead of GG) [20], pcDNA3.1-HA--FAT10 [66], pcDNA3.1-HA-FAT10-AV [20], pcDNA6.1-FAT10 [57], pcDNA6.1-FAT10-AV were used. pcDNA6.1-FAT10-AV was generated by site-directed mutagenesis using primers

AA-394 5`-GCGGCCGCGTCACACTGCAATACAATAAGATGCCAG-3` and AA-395 5`-
CTGGCATCTTATTGTATTGCAGTGTGACGCGGCCGC-3` and pcDNA6.1-FAT10 as tem-
plate. For expression of differently tagged, N-terminally truncated HUWE1 variants,
pcDNA3-6His-HUWE1ΔN, pcDNA3-HA-HUWE1ΔN [30] or its active site cysteine mutants
pcDNA3-HA-HUWE1ΔN-C/S (C4341S) [30] and pcDNA3-HA-HUWE1ΔN-C/A (C4341A)
were used. For expression of shorter HUWE1 variants, pCMV-6His-HECT as well as the cys-
teine mutants pCMV-6His-HECT-C/A (C4341A), pCMV-6His-HECT TM (C4099A,
C4184A, C4367A) and pCMV-6His-HECT QM (C4099A, C4184A, C4341A, C4367A) were
used. To generate pCMV-6His-HECT as well as pCMV-6His-HECT TM and pCMV-6His-
HECT QM, DNA of the HECT variants was amplified from the corresponding pSKB2-6His-
3C-HUWE1-HECT, -TM or -QM constructs via PCR with the following primers: 5'-EcoRI-
6his-HECT 5'-CCGGAATTCATGGGCAGCAGCCATCATC-3' and 3'-BglII6his-HECT 5'-
GGAAGATCTTTAGGCCAGCCCAAAGCCTTC-3'. Then, the desired HECT variant was
inserted into pCMV-FLAG (C-terminal tag, clontech/Takara) using the restriction sites EcoRI
and BglII. HA-tagged ubiquitin was expressed from pcDNA3.1-HA-Ub (gift from M. Basler,
University of Konstanz, Germany). pCMV6-AMBRA1-Myc-DDK (OriGene (NM_017749)
#RC206255) was used for expression of FLAG-tagged AMBRA1. For the generation of
pcDNA3-HA-HUWE1ΔN-C/A, pGEX-4T-GST-HectH9trunc-C/A, pSKB2-6His-
3C-HUWE1-HECT QM as well as pCMV-6His-HECT-C/A a site directed mutagenesis of the
active site cysteine C4341 to alanine was performed with pcDNA3-HA-HUWE1ΔN, pGEX-
4T-GST-HectH9trunc, pSKB2-6His-3C-HUWE1-HECT TM or pCMV-6His-HECT as tem-
plate, respectively, and the following primers: PR4-41 5'-CTGCCTTCAGCTCACACAG
CCTTTAATCAGCTGGATCTG-3' and PR4-42 5'-CAGATCCAGCTGATTAAAGGCTGTG
TGAGCTGAAGGCAG-3'. The HA-tag of HUWE1ΔN was exchanged to pcDNA3-6His-
HUWE1ΔN with the primers SM-1 5'-GAATTCCTCGACGGATCATCGAATTCACCA
TGCATCATCATCATCATCATGCTAGCGGATCCATGAACGCTTCTCCC-3' and i-SM-1 5'-
GGGAGAAGCGTTCATGGATCCGCTAGCATGATGATGATGATGATGCATGGTGAATTCG
ATGATCCGTCGAGGAATTC-3'.

All HUWE1 constructs used are schematically shown in Fig 1A. The sequences of all gener-
ated plasmids were verified by sequencing (Microsynth AG).

## Induction of endogenous FAT10 expression

Expression of endogenous FAT10 was induced by treating HEK293 cells with a combination
of the two pro-inflammatory cytokines tumor necrosis factor-α (TNF, 600 U/mL) and inter-
feron-γ (IFN-γ, 300 U/mL) (both from Peprotech GmbH) [67], as described previously [67].

## Immunoprecipitation, SDS-PAGE and Western blot

For transient protein expression, HEK293 cells were transfected with different expression con-
structs using the TransIT LT1 Transfection Reagent (Mirus Bio LLC). After 24 hours, cells
were lysed for 30–60 min on ice in lysis buffer containing 20 mM Tris/HCl (pH 7.6), 50 mM
NaCl, 10 mM MgCl2 and 1% Nonidet P-40 with 1x protease inhibitor mix (cOmplete™ mini
EDTA-free protease inhibitor cocktail, Roche). After taking a sample for the loading control,
proteins were immunoprecipitated from the cleared lysate for at least 2 hours at 4°C, as indi-
cated, with monoclonal anti-HA agarose antibody HA-7 (Sigma Aldrich) or Anti-FLAG M2
Affinity Gel (Sigma Aldrich). For immunoprecipitation of endogenous FAT10 protein A
sepharose (Sigma Aldrich) in combination with monoclonal mouse FAT10 antibody 4F1 ([20]
and Enzo Lifesciences) was used. Samples were washed as described before [68] and then
boiled in 5x SDS gel sample buffer. For detection under reducing or non-reducing conditions,

samples were supplemented with 5x gel sample buffer with or without 4% β-ME, respectively, and boiled. Proteins were either separated on 4–12% NuPAGE Bis-Tris SDS gradient gels (Invitrogen) or on 12.5% or 6% Laemmli SDS gels. Subsequently, proteins were transferred onto a Protran nitrocellulose membrane (Sigma) and Western blot analysis was performed using the following antibodies: anti-ARF-BP1 (rabbit, polyclonal, Sigma Aldrich), anti-HUWE1/Mule (rabbit, polyclonal, Abcam), anti-FAT10 (rabbit, polyclonal) [18], anti-USE1 (rabbit, polyclonal) [20], anti-β-actin (mouse, monoclonal, clone AC-15, Abcam,), anti-FLAG-HRP (mouse, monoclonal, clone M2, Sigma Aldrich), anti-HA-POX (mouse, monoclonal, clone HA-7, Sigma Aldrich), anti-6His-POX (mouse, monoclonal, clone His-1, Sigma Aldrich), anti-mouse-HRP (goat, polyclonal, Jackson Immuno Research), anti-rabbit-HRP (goat, polyclonal, Jackson Immuno Research). For quantification ECL signals were analysed with densitometry calculations (ImageLab Software, BioRad, Basel, Switzerland) and the values normalized to the respective proteins in the lysate or to the loading control β-actin.

## Generation of knockout cells

To generate HUWE1-knockout cells, HEK293 cells were transfected with a CRISPR gRNA plasmid expressing Cas9, a target sequence (HS0000480602,) and GFP (p01-U6-gRNA: CMV-Cas9-2A-tGFP, Merck) to generate HUWE1-deficient cells by CRISPR-Cas9-mediated gene inactivation. After 24 hours, the cells were sorted for GFP-positive single cells by fluorescence activated cell sorting (FACS). The clones were expanded and screened for HUWE1 knockout by Western blot analysis with the anti-HUWE1 (Mule) antibody.

## Recombinant protein expression and purification

For *in vitro* experiments, recombinant FAT10 and FAT10-AV as well as 6His-USE1 were expressed and purified as described earlier [20, 63, 67]. For purification of GST-tagged HectH9 and HectH9-C/A, E. coli BL21 (DE3) were transformed with pGEX-4T-GST-HectH9-trunc [30] (kindly provided by M. Scheffner, University of Konstanz, Germany) and pGEX-4T-GST-HectH9trunc-C/A expression constructs and grown at 37˚C in modified LB medium (HSG medium, 13.5 g/L peptone, 7 g/L yeast extract, 15 g/L glycerol, 2.5 g/L NaCl, 2.3 g/L K2HPO4, 1.5 g/L KH2PO4, 0.14 g/L MgSO4x7H2O, 90 ˚L/L antifoam, pH 6.8, supplemented with 100 ˚g/mL ampicillin). At OD600 of about 0.6 protein expression was induced upon addition of 0.5 mM IPTG and cells were cultured at 30˚C overnight. Cells were pelleted and mechanically lysed in lysis buffer (PBS, 1 mM TCEP, 1 ˚M PMSF, supplemented with 1x protease inhibitor mix) with at least 2 cycles at 2.5 kbar in a cell disrupter (Constant Cell Disrupter TS, Constant Systems Ltd.). Glutathione Sepharose beads (GE Healthcare) pre-equilibrated in lysis buffer were added to the cleared lysates and incubated rolling at 4˚C for 90 minutes. Beads were washed four times with washing buffer (PBS, 0.1% Triton X-100, 1 mM TCEP) and protein was eluted in fractions with elution buffer (25 mM Tris/HCl pH 8.0, 1 mM TCEP, 10 mM glutathione red.). Finally, buffer was exchanged to storage buffer (25 mM Tris/HCl pH 8.0, 1 mM TCEP) using PD-10 Desalting Columns containing Sephadex G-25 medium (GE Healthcare) and protein concentration as well as purity determined with BCA assays and Western blot staining with Ponceau as well as anti-GST antibodies (mouse, monoclonal, clone B 14, Santa Cruz), respectively.

His-tagged HUWE1-HECT protein variants were expressed in E. coli BL21 (DE3)-RIPL using the pET-28a derived expression constructs pSKB2-6His-3C-HUWE1-HECT [36], pSKB2-6His-3C-HUWE1-HECT-C/A [36], pSKB2-6His-3C-HUWE1-HECT TM [36], and pSKB2-6His-3C-HUWE1-HECT QM. As previously described, bacteria were grown at 37˚C in modified LB medium (supplemented with 100 ˚g/mL kanamycin). Protein expression was

induced with 0.5 mM IPTG at 21˚C overnight before cells were harvested and mechanically lysed in binding buffer (80 mM HEPES, 0.5 mM NaCl, 1 mM TCEP, 10% glycerol, 10 mM imidazole, pH 7.0, supplemented with 1x protease inhibitor mix), as described above. Where necessary, DNase I digest for 30 min on ice combined with sonication was performed. Cleared lysates were loaded onto a HisTrap column (HisTrap FF 5mL, Cytiva) to per-form affinity chromatography using the AektaPure system (Cytiva, formerly GE Healthcare). His-tagged proteins were eluted by the stepwise addition of buffer containing 1 M imidazole. Fractions containing the protein of interest were combined and concentrated before they were subjected to a size exclusion chromatography (SEC, Hi-Load 16/60 Superdex 75pg, Cytiva) using the AektaPure system with gel filtration buffer (20 mM HEPES, 150 mM NaCl, 1 mM EDTA, 1 mM TCEP, pH 7.0). Fractions containing the protein of interest were determined by Coomassie stained SDS gels and Western blotting, and protein concentration was measured with a NanoDrop spectrophotometer. Purified recombinant proteins were stored at -80˚C.

### *In vitro* FAT10ylation assay

*Semi-in vitro* and *in vitro* FAT10ylation assays were performed in a final volume of 20 ˚L with 1x *in vitro* buffer (20 mM Tris/HCl (pH 7.6), 50 mM NaCl, 10 mM MgCl2, 0.1 mM DTT, 1x protease inhibitor mix (Roche), with or without 4 mM ATP) and the proteins indicated. If the sample was analyzed under both, non-reducing and reducing conditions, the reaction was performed in a final volume of 40 ˚L, respectively. The samples were incubated shaking at 37˚C for 30 min. The reaction was stopped by adding 5 x SDS gel sample buffer before samples were subjected to subsequent SDS-PAGE and Western blot analysis as described above. For *semi-in vitro* assays HA-tagged HUWE1ΔN and HUWE1ΔN-C/A were immunoprecipitated with monoclonal anti-HA agarose antibody HA-7 (Sigma Aldrich) from lysates of transiently transfected HEK293 HUWE1-KO cells. Immunoprecipitated proteins were washed extensively with NET-TN, NET-T buffer [68] and 1x *in vitro* buffer twice before using them for *in vitro* FAT10ylation reactions.

The following amounts of recombinant proteins were used for the *in vitro* reaction: 0.15 ˚g FLAG-UBA6 (Enzo Life Sciences), 0.7 ˚g 6His-USE1, 2.7 ˚g FAT10 or FAT10-AV, 1.5 ˚g ubiquitin (Boston Biochem), 2.5 ˚g GST-HectH9 or GST-HectH9-C/A and 1.8 ˚g 6His-HECT, 6His-HECT-C/A, 6His-HECT TM or 6His-HECT QM.

### Cycloheximide chase assay

For cycloheximide chase experiments, transfected cells were treated for the indicated time periods with cycloheximide (CHX, 50 ˚g/mL final concentration, Sigma) or the same amount of DMSO as control before cell lysis. Where indicated, the proteasome was inhibited six hours prior to harvesting with MG132 (10 ˚M final concentration, Enzo Life Sciences). The cell lysates were then proceeded as described above.

### HUWE1 inhibition

The HUWE1 inhibitors BI8622 and BI8626 [59] (stock concentrations 10 mM in DMSO, kindly provided by S. Lorenz, Max Planck Institute for Multidisciplinary Sciences, Göttingen, Germany) were used at a final concentration of 10 ˚M. Cells were treated with the inhibitors for six hours before harvesting and lysis.

## Supporting information

**S1 File.**
(PDF)

**S1 Raw images.**
(PDF)

## Acknowledgments

We gratefully acknowledge Prof. Dr. Marcus Groettrup, who sadly passed away during the course of this study. We thank Dr. Sonja Lorenz for the kind contribution of plasmids and inhibitors, Prof. Dr. Martin Scheffner for providing a plasmid and Dr. Edith Uetz-von Allmen for single cell sorting.

## Author Contributions

**Conceptualization:** Stefanie Mueller, Annette Aichem.

**Formal analysis:** Stefanie Mueller.

**Funding acquisition:** Annette Aichem.

**Investigation:** Stefanie Mueller, Johanna Bialas, Stella Ryu, Nicola Catone.

**Methodology:** Stefanie Mueller, Annette Aichem.

**Project administration:** Annette Aichem.

**Supervision:** Annette Aichem.

**Writing – original draft:** Stefanie Mueller.

**Writing – review & editing:** Annette Aichem.

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
