## [Decision Letter · Decision Letter 0]

26 Jun 2023

PONE-D-23-12523The ubiquitin-like modifier FAT10 covalently modifies HUWE1 and strengthens the interaction of AMBRA1 and HUWE1PLOS ONE

Dear Dr. Aichem,

Thank you for submitting your manuscript to PLOS ONE. After careful consideration, we feel that it has merit but does not fully meet PLOS ONE’s publication criteria as it currently stands. Therefore, we invite you to submit a revised version of the manuscript that addresses the points raised during the review process.

Both reviewers found your manuscript interesting.  The experiments were carefully designed with inclusion of necessary and stringent controls. The results were clearly described, adequately discussed, and in general support the conclusions. Overall, the manuscript is well written and easy to follow.

Please address the minor concerns including Fig. 1b, the 2nd FLAG blot, line 4-5, if the signals suggest a non-covalent interaction between FAT10 and HUWE1, why would these signals remain detectable under reducing conditions (the same blot, lane 9-10)?

We look forward to receiving your revised manuscript.

Kind regards,

Aldrin V. Gomes, Ph.D.

Academic Editor

PLOS ONE

Reviewers' comments:

Reviewer's Responses to Questions

**Comments to the Author**

1. Is the manuscript technically sound, and do the data support the conclusions?

Reviewer #1: Yes

Reviewer #2: Yes

2. Has the statistical analysis been performed appropriately and rigorously? 

Reviewer #1: N/A

Reviewer #2: Yes

3. Have the authors made all data underlying the findings in their manuscript fully available?

Reviewer #1: Yes

Reviewer #2: Yes

4. Is the manuscript presented in an intelligible fashion and written in standard English?

Reviewer #1: Yes

Reviewer #2: Yes

5. Review Comments to the Author

Reviewer #1: This is an interesting study that for the first time identfies HUWE1 as a putative FAT10 modified target. The experiments were carefully designed with inclusion of necessary and stringent controls. The results were clearly described, adequately discussed, and in general support the conclusions. Overall, the manuscript is well written and easy to follow.

There is only one minor question:

Fig. 1b, the 2nd FLAG blot, line 4-5, if the signals suggest a non-covalent interaction between FAT10 and HUWE1, why would these signals remain detectable under reducing conditions (the same blot, lane 9-10)?

Reviewer #2: I found the manuscript to be detailed and thoughtful. The authors have made all data available, the experimentation and the statistical analysis is thorough, and the manuscript is well-written. This manuscript adheres to the submission criteria of PLOS ONE.

6. PLOS authors have the option to publish the peer review history of their article (what does this mean?). If published, this will include your full peer review and any attached files.

Reviewer #1: No

Reviewer #2: No

---

## [Author Response · Author response to Decision Letter 0]

3 Jul 2023

Point-by-point reply for manuscript #PONE-D-23-12523 by Mueller et al.

We would like to thank our reviewers for their positive comments.

Reviewer #1: This is an interesting study that for the first time identfies HUWE1 as a putative FAT10 modified target. The experiments were carefully designed with inclusion of necessary and stringent controls. The results were clearly described, adequately discussed, and in general support the conclusions. Overall, the manuscript is well written and easy to follow.

There is only one minor question:

Fig. 1b, the 2nd FLAG blot, line 4-5, if the signals suggest a non-covalent interaction between FAT10 and HUWE1, why would these signals remain detectable under reducing conditions (the same blot, lane 9-10)?

We thank our reviewer for this question. In Fig 1b, we have performed an immunoprecipitation and analyzed the proteins both, under non-reducing, and reducing conditions. Under non-reducing conditions, proteins immunoprecipitated from cell lysates are denatured by SDS, which disrupts non-covalent bonds such as hydrogen bonds, hydrophobic interactions or ionic bonds within and between proteins. The addition of β-ME to the SDS PAGE sample buffer as reducing agent does not only break thioester bonds between thiol groups of cysteines and glycine residues, but also disulfide bonds between cysteine residues. This is different to an immunoprecipitation which is performed under denaturing conditions. In this case, cells are already lysed under denaturing conditions (e.g. in the presence SDS) which dissolve all non-covalent interactions before the immunoprecipitation is performed. 

In the experiment shown in Fig 1b, cells were lysed and cleared lysates were subjected to an immunoprecipitation using HA-reactive agarose coupled antibodies. After washing the beads with washing buffer, 5x SDS gel sample buffer (without β-ME) was added directly to the beads and the beads were boiled (non-reducing conditions). Then, the supernatant was split equally and β-ME (4% final concentration) was directly added to one of the two portions and boiled again (reducing conditions). Therefore, all proteins, which were bound to the immunoprecipitated HA-tagged HUWE1 were still in the suspension what explains why non-covalently interacting FAT10 was detectable under both conditions. 

We have now added the following explanations to the text:

Lines 107-109:

“Moreover, also a non-covalent interaction of FAT10 and HUWE1 was detected, both, under non-reducing and reducing (4% �-ME) conditions (Fig 1b, second panel, IP: HA, IB: FLAG, lanes 4-5, and 9-10).”

Lines 163-167:

Samples were analyzed under non-reducing and reducing (4 % β-ME) conditions by SDS-PAGE. For this purpose, immunoprecipitated samples were first boiled in 5x SDS PAGE sample buffer without β-ME, and then split into two portions. While one portion was left untreated (non-reducing conditions), 4% β-ME (final concentration) was added to the second portion and boiled again (reducing conditions).

Reviewer #2: I found the manuscript to be detailed and thoughtful. The authors have made all data available, the experimentation and the statistical analysis is thorough, and the manuscript is well-written. This manuscript adheres to the submission criteria of PLOS ONE.

We would like to thank our reviewer for the positive evaluation of our manuscript.

---

## [Decision Letter · Decision Letter 1]

1 Aug 2023

The ubiquitin-like modifier FAT10 covalently modifies HUWE1 and strengthens the interaction of AMBRA1 and HUWE1

PONE-D-23-12523R1

Dear Dr. Aichem,

We’re pleased to inform you that your manuscript has been judged scientifically suitable for publication and will be formally accepted for publication once it meets all outstanding technical requirements.

Kind regards,

Aldrin V. Gomes, Ph.D.

Academic Editor

PLOS ONE

Additional Editor Comments (optional):

Reviewers' comments:

Reviewer's Responses to Questions

**Comments to the Author**

1. If the authors have adequately addressed your comments raised in a previous round of review and you feel that this manuscript is now acceptable for publication, you may indicate that here to bypass the “Comments to the Author” section, enter your conflict of interest statement in the “Confidential to Editor” section, and submit your "Accept" recommendation.

Reviewer #1: All comments have been addressed

2. Is the manuscript technically sound, and do the data support the conclusions?

Reviewer #1: Yes

3. Has the statistical analysis been performed appropriately and rigorously? 

Reviewer #1: Yes

4. Have the authors made all data underlying the findings in their manuscript fully available?

Reviewer #1: Yes

5. Is the manuscript presented in an intelligible fashion and written in standard English?

Reviewer #1: Yes

6. Review Comments to the Author

Reviewer #1: I have no further comments. The manuscript is now acceptable for publicaiton in PLOS one.

7. PLOS authors have the option to publish the peer review history of their article (what does this mean?). If published, this will include your full peer review and any attached files.

Reviewer #1: No

---

## [Editor Report · Acceptance letter]

3 Aug 2023

PONE-D-23-12523R1 

The ubiquitin-like modifier FAT10 covalently modifies HUWE1 and strengthens the interaction of AMBRA1 and HUWE1 

Dear Dr. Aichem:

I'm pleased to inform you that your manuscript has been deemed suitable for publication in PLOS ONE. Congratulations! Your manuscript is now with our production department. 

Kind regards, 

on behalf of

Dr. Aldrin V. Gomes 

Academic Editor

PLOS ONE